# Wear Study of Coated Mills during Circumferential Milling of Carbon Fiber-Reinforced Composites and Their Influence on the Sustainable Quality of the Machined Surface

**Tomáš Knápek** * [ID], **Štěpánka Dvořáčková** [ID] **and Artur Knap** [ID]

Assembly and Engineering Metrology, Department of Machining, Faculty of Mechanical Engineering, Technical University of Liberec, 461 17 Liberec, Czech Republic
* Correspondence: tomas.knapek@tul.cz

**Abstract:** Composite materials made of fiber-reinforced plastic laminates are highly susceptible to surface damage caused by wear during contour milling, especially with inappropriate tool and cutting material properties. Improper choice of tools and cutting conditions lead to delamination between applied layers, thermal damage of materials in the polymer matrix, and reduction of the edge quality of cutting tools. The study was devoted to circumferential milling of twill-bonded CFRP (carbon-fiber-reinforced polymer) sheets with a focus on cutting forces and tool flank face wear, including their effect on the machined surface structure, roughness, and topography of the laminate. The main objective of the study is to investigate the feasibility of applying conventional coated tools, which are not primarily designed for milling CFRP, in comparison to a dedicated DLC (diamond-like carbon) coated tool, due to economic and distribution availability and the possibility of providing suitable cutting conditions during milling. The study provides results confirming the possibility of using conventional tools for machining CFRP and provides relevant experimental results that can be implemented for optimal tool selection, tool life criteria, cutting conditions, and machining strategies including low energy consumption. The best values of the investigated parameters were obtained when using the ECSSF (instrument designation) tool with DLC coating.

**Keywords:** milling; composite systems; carbon fibers; tool wear; tool parameters





## 1. Introduction

Composite materials with a matrix based on resin or polymer belong to a group of materials with specific properties, the use of which is constantly increasing along with the requirements for tools [1]. The mutual combination of matrix and reinforcing fibers can achieve different mechanical properties, which must be considered when choosing a cutting tool [2,3].

In addition to the tool itself, the cutting conditions also significantly influence the quality of machining composite materials. It depends on the correctly selected feed rate, on the revolutions, and on ensuring a perfect cut, that is, on preventing the tool from rubbing against the surface of the workpiece. The material delaminates or the fibers break if the tool does not cut correctly [4–6].

Machining composite materials is different and challenging. The choice of suitable cutting tools and cutting conditions requires experience in chip machining [7,8].

From the point of view of the issue of machining composite materials with carbon fibers, it is necessary that the cutting tool actually cuts the composite including the reinforcement fibers [9]. There must be no breaking or pulling of the fibers from the matrix and fraying of the edge of the machined surface. The limit temperature must not be exceeded in the cutting zone or in its surroundings, at which thermal degradation of the matrix occurs [10]. The edge of the cutting tool must be sharp, with a smooth surface with a low coefficient of friction, so that there is no unwanted increase in temperature at the point of

the cut. The heat from the cutting site should be dissipated primarily by the tool (therefore, tools made of cemented carbide are preferable due to their higher thermal conductivity compared to high-speed steel) [11,12]. The separated material has a crumbly nature and not a chip shape, therefore the tools must have sufficient chip gaps to prevent stagnation of the separated material in the work zone, which could increase its temperature [13,14]. The component of the cutting force acting perpendicularly to the individual layers of the composite should not cause the separation of these layers, the so-called delamination, which is, together with fiber breakage, the most common type of damage [15,16].

Using coated cutting tools for CFRP (carbon-fiber-reinforced polymer) composites reduces tool wear. With the development of manufacturing technology, the use of coated tools has increased, and today approximately 80% of all machining operations are carried out with coated cutting tools [14,17].

Severe mechanical wear at the cutting tool is one of the main issues in machining CFRP and is primarily responsible for limited tool life. Progressive tool wear is associated with a continuously changing active micro-geometry, which affects the tool/material interaction in the contact zone and thus the resulting process forces and the tool performance [18,19].

In the case of a composite with long, unidirectionally oriented fibers, it is advisable to consider the direction of the cutting force with regard to the orientation of the fibers, so that the eventual failure is directed to places that will be removed by further processing [20,21]. The dominant mode of wear of cutting tools is abrasion, given the high hardness of the fibers; the thermal loading of the tool initiates oxidation and diffusion wear mechanisms, while the effect of adhesion is negligible. There have been many studies on the investigated issue, but each study is always specific in terms of the cutting tools used, cutting conditions, the method of measuring selected parameters, and the construction of a composite with CFRP. Research is still up-to-date and every new information can lead to new knowledge in the given area.

The present study was devoted to circumferential milling of twill bonded CFRP sheets with a focus on cutting forces and tool flank face wear, including their effect on the machined surface structure, roughness, and topography of the laminate. For the study, three types of tools were selected for different uses—for laminate composite material, non-ferrous metals, and aluminum, and for steel, stainless steel, and cast iron.

Part of the study was to understand the nature of the phenomena (delamination between laminate layers, fiber pulling, tool wear, etc.), accompanying the milling process of composites with carbon fibers in connection with the behavior of the tool, whether directly intended for machining composites or not. The study provides relevant experimental results that can be implemented for optimal tool selection, tool life criteria, cutting conditions, and machining strategies, including low energy consumption.

## 2. Materials and Methods

Glossy laminated 3 K CFRP plates with a thickness of 1 and 3 mm were used for the experimental study. Used composite samples were purchased from the manufacturer Kavan Rc (Doubravice, Czech Republic). The 1 mm thick laminate consists of 4 woven layers of twill weave. The 3 mm thick laminate consists of 12 woven layers of twill weave. Technical data given by the manufacturer are in Table 1. A section of the purchased laminate (3 mm board) is shown in Figure 1. The composition structure shown is determined by the manufacturer.

**Table 1.** Material characteristics of the examined CFRP laminate.

| Resin Type | Epoxy |
|---|---|
| Carbon fiber type | HT (high tenacity) |
| Weave type | Plain 2 × 2 |
| Number of filaments per roving | 3 K |
| Fiber volume friction | 40% |
| Number of plies | 4, 12 |
| Ply thickness in laminate | 1 mm, 3 mm |
| Density | 1850 kg/m$^3$, 1670 kg/m$^3$ |

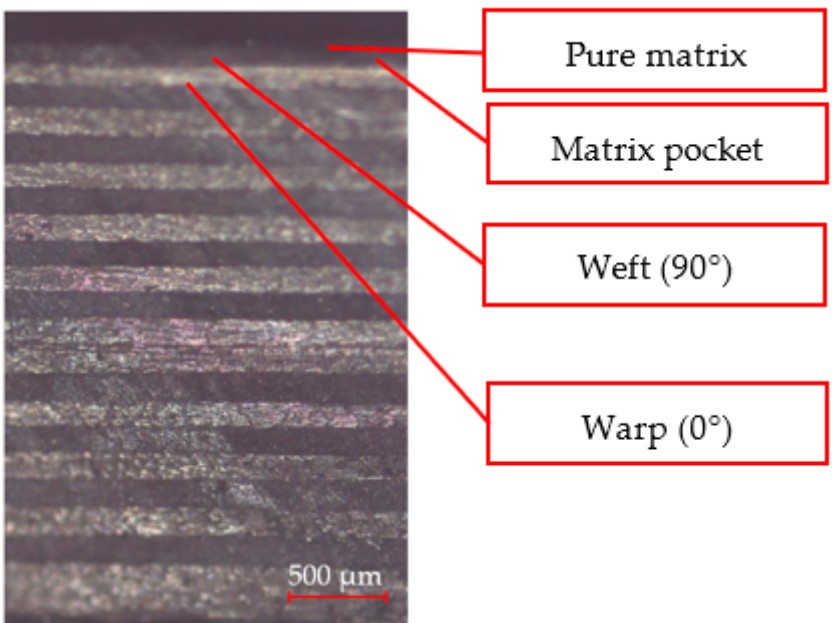

**Figure 1.** Cut through 3 mm thick laminate.

The surface of the laminate consists of a thin layer of pure resin with a thickness of 10–15 μm. The thickness of the resin layer changes as the warp and weft yarns are placed. Next comes the weft yarn. Then, there is the warp yarn below and again the weft yarn. The undulation path of the warp and weft yarns in a twill weave can be approximately described by a sine function. A grouping of matrix material is visible between the warp and weft yarns.

Laminated CFRP plates with a thickness of 1 and 3 mm were cut in the form of plates with dimensions of 402 mm × 250 mm. The boards were adjusted/cut to sample dimensions of 200 mm × 250 mm using a band saw. The length of the milled edge of 200 mm was chosen for the experiment.

For the study were used 3 types of cutters, see Figure 2. General and material properties are given in Tables 2 and 3.

**Table 2.** General properties of tools.

| Tool | Coating | Coating Thickness [μm] | L1 [mm] | L [mm] | D [mm] | Helix Angle | Teeth |
|---|---|---|---|---|---|---|---|
| Finishing cutter ECSSF | DLC | 4 | 15 | 60 | 6 | 8° | 6 |
| Carbide cutter A100 | CrN | 4 | 15 | 50 | 6 | 30° | 2 |
| Universal cutter G550 | UNICO | 4 | 15 | 50 | 6 | 45° | 2 |

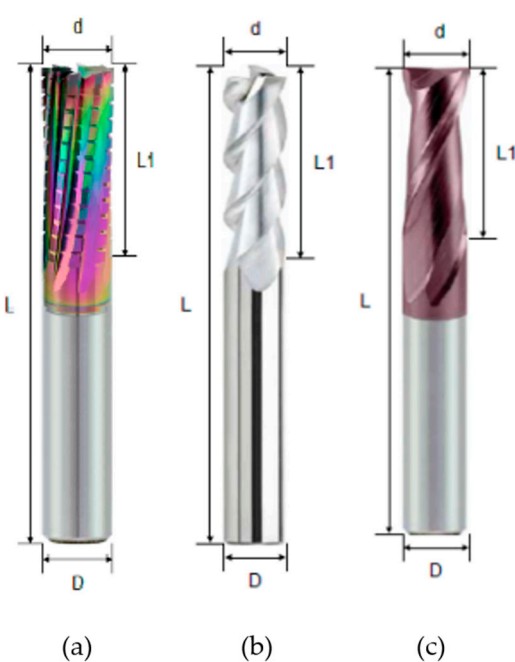

(a)        (b)        (c)

**Figure 2.** Tools used for the study. (**a**) ECSSF milling cutter; (**b**) A100 milling cutter; (**c**) G550 milling cutter.

**Table 3.** Material properties of tools.

| Tool | Application Group ISO | SC Type | Chemical Composition [%] | | | Medium Grain Size [μm] | Density [g/cm$^3$] | Flexural Strength [MPa] | Hardness HV |
|------|------|------|------|------|------|------|------|------|------|
| | | | WC | TiC + TaC + NbC | Co | | | | |
| Finishing cutter ECSSF | K10 | H10 | 94 | - | 6 | 1–2 | 14.8 | 1800 | 1600 |
| Carbide cutter A100 | K30 | H30 | 91 | - | 9 | 2 | 14.6 | 2000 | 1380 |
| Universal cutter G550 | M10 | U105 | 84.8 | 9.7 | 5.5 | 1–2 | 13.2 | 1700 | 1600 |

Used tools were purchased from the manufacturer Winstar (Tainan, Taiwan). The tools were of the same diameter, but with different geometry, different coating and in the case of the ECSSF cutter, also a different number of teeth, due to the investigation of different accompanying and subsequent phenomena during milling with these tools.

The ECSSF finishing cutter is designed for CFRP/GFRP laminate composite materials, the microhardness of DLC surfaces is between 3300 HV0.05 and 4200 HV0.05. The carbide cutter A100 is intended for machining non-ferrous metals, aluminum, and aluminum alloys, the microhardness of the CrN coating is between 1200 HV0.25 and 2900 HV0.25. The universal milling cutter G550 is suitable for steel, stainless steel, cast iron, and hardened material up to 55 HRC, the hardness of the UNICO surface layer is 2800 HV0.05 and 3100 HV0.05.

Milling was carried out on a 3-axis DMG MORI CMX 600 (DMG Mori Seiki, Nagoja, Japan) milling center with a spindle power of 13 kW and a maximum rotation speed of 12,000 rpm. The instruments were clamped using a heat clamp. To minimize the influence of vibrations, the samples were fixed over a large area to the clamping device and the clamping device to the KISTLER 9265 B (Kistler Instrument Corp, Amherst, NY, USA) dynamometer, see Figure 3. To reduce the dust particles of the chips, an auxiliary air extraction was installed at the cutting site using an industrial vacuum cleaner.

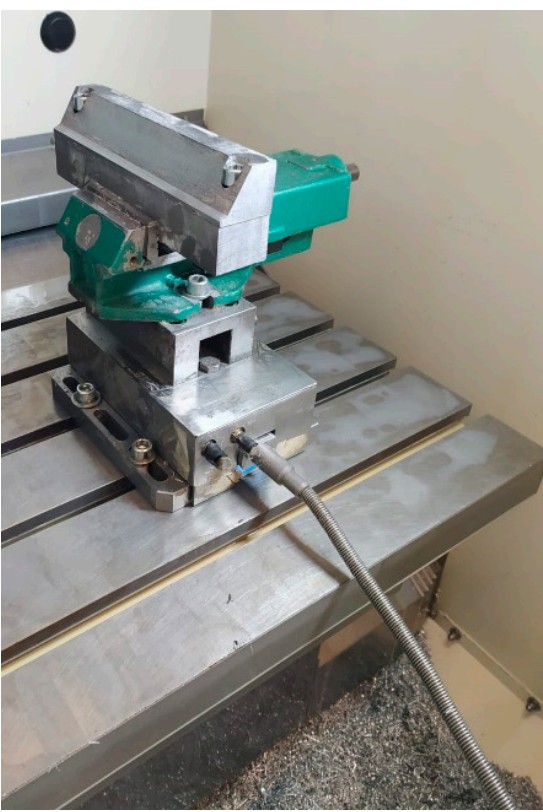

**Figure 3.** Fixture for mounting CFRP sample on the experimental device—piezoelectric dynamometer KISTLER 9265 B.

As part of the study, circumferential down milling without process fluid was implemented. A 200 mm long board edge was milled. The length of the tool tooth path was determined according to Equations (1) and (2), Figure 4, and Table 4.

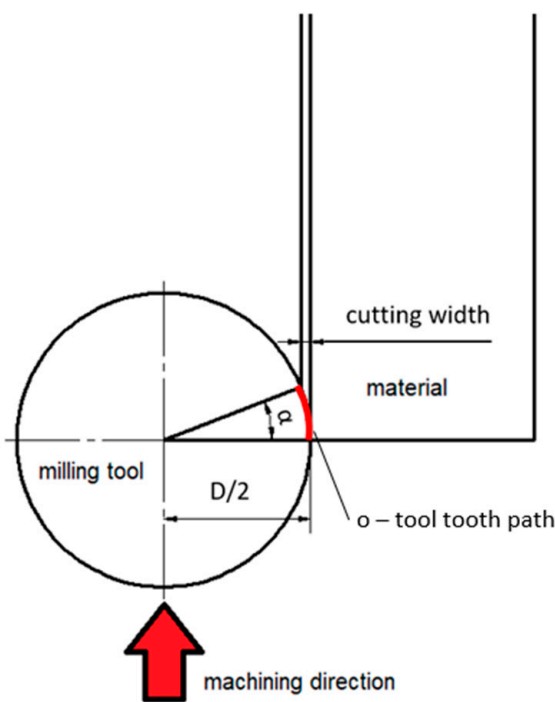

**Figure 4.** Scheme for calculating the tool tooth path.

**Table 4.** Used cutting conditions.

| Cutting Condition | Value |
|---|---|
| Tool diameter D [mm] | 6 |
| Cutting speed [m/min] | 200 |
| Feed per tooth $f_z$ [mm] | 0.02 |
| Cutting width [mm] | 0.2 |

Circumference of the circular section of the tool tooth in the cut:

$$o = \frac{\pi}{180} \times \alpha \times \frac{D}{2} = \frac{\pi}{180} \times 21 \times \frac{6}{2} = 1.1 \text{ mm}. \tag{1}$$

Traveled tool tooth path after one section:

$$s = \frac{o \times L}{f_z} = \frac{1.1 \times 200}{0.02} = 11000 \text{ mm} = 11 \text{ m}. \tag{2}$$

The criterion for the evaluation of the results was chosen to be the distance traveled by the tool tooth. For the purpose of the experiment, 15 milling paths were selected. According to Equation (2), the length of one tool tooth traveled is 11 m. Therefore, the total distance traveled by the tool tooth in one control section is 165 m ($15 \times 11$ m). A total of 4 control sections ($5\times$, $10\times$, $15\times$, $20 \times 165$ m) were investigated.

The laminated CFRP plate specimens were always clamped so that they protruded 4 mm beyond the edge of the clamping device. Cutting conditions were chosen for milling, see Table 4, which were selected according to the recommendations of the tool manufacturer.

Each tool machined 2 different material thicknesses, i.e., 1 and 3 mm, since the material thickness was not large, one tool could be used for both material thicknesses due to the displacement of the tool in the Z axis so that the previous wear of the tool did not interfere with the still unworn part of the tool.

The force measurement was carried out using a three-component piezoelectric dynamometer KISTLER 9265 B. The force measurement was carried out for each cycle of milling and tool travel of 165 m. A MITUTOYO SV-2000N2 SURFTEST (Mitutoyo, Kanagawa, Japan) contact profilometer was used to measure surface roughness. Control of the device and evaluation of the results was realized by the Surfpak (v.12.2, 2004, Mitutoyo, Kanagawa, Japan) software. Three roughness parameters were evaluated, namely $R_a$, $R_z$, and $R_t$, which correspond to ISO, DIN, ANSI, and JIS standards. The roughness measurement was carried out along the traveled path of $5 \times 165$ m. Until the traveled path of 3300 m ($20 \times 165$ m), the roughness measurement on the material was carried out 4 times (after every 5 repetitions) to determine the dependence between the wear of the tool and the roughness of the surface of the material.

Sample delamination measurements, tool wear (VB), and laminate analysis were performed using a KEYENCE VK-X1100 3D (Keyence, Itasca, IL, USA) laser non-contact profilometer. Control of the device and evaluation of the results was realized by the MultiFileAnalyser (VK-H1XMD, 2019, Keyence, Itasca, IL, USA) software.

Statistical data processing—the arithmetic mean $\bar{x}$ was always calculated from the measured data, then the measurement uncertainty was calculated. The measurement uncertainty was determined in accordance with document EA-4/02 M:2013. The individual measurement uncertainties listed for the results were calculated according to the EA document mentioned above, where the type A uncertainty and then the type B uncertainty were calculated. The instrument/machine uncertainty and other equipment and gauges accounted for the largest proportion. Subsequently, the combined uncertainty uc was calculated, from which the resulting expanded uncertainty was calculated (expansion k = 2).

## 3. Results

The study was focused on the milling of CFRP plates with twill weave, especially on:

- Delamination of fibers and damage to the machined material (laminate) after milling;
- Cutting forces depending on tool wear;
- Roughness parameters of the machined surface depending on tool wear;
- Overall comparison of the resulting wear of individual tools.

### 3.1. Delamination of Fibers and Damage to the Machined Material (Laminate) after Milling

The study focused on fiber delamination and damage to the machined material after milling confirming some conclusions according to [6]. Publication [16] follows from publication [22–25].

According to [22], delamination occurs in the upper layers during the cutting action of the milling tool, which bends the fibers outwards and deviates them from the plane of the laminate. This induces tension between the layers of the laminate, causing the layers to separate. The authors [23] divided the resulting delamination into different types. Type I delamination describes the surface breaking of fiber bundles on the machined surface (the edge of the machined surface), which takes place in the plane of the top layer. Type II delamination is characterized by fibers that protrude beyond the machined edge without causing noticeable damage to the surface.

According to [24], a mixed form of type I and II delaminations can also be found here. This type contains bundles of fibers protruding beyond the machined edge accompanied by significant surface damage to the material. Which of the respective types occurs depends to a large extent on the orientation of the fibers in the top layer.

According to [25], delamination is induced at the point of initial contact of the fiber with the tool, where the first cut of the fiber occurs. It follows that the maximum length of the protruding fiber is the distance between the initial contact and the edge of the machined (milled) surface in the longitudinal direction of the fiber. The milling path was oriented at a 90° angle to the weft yarn, Figures 5 and 6.

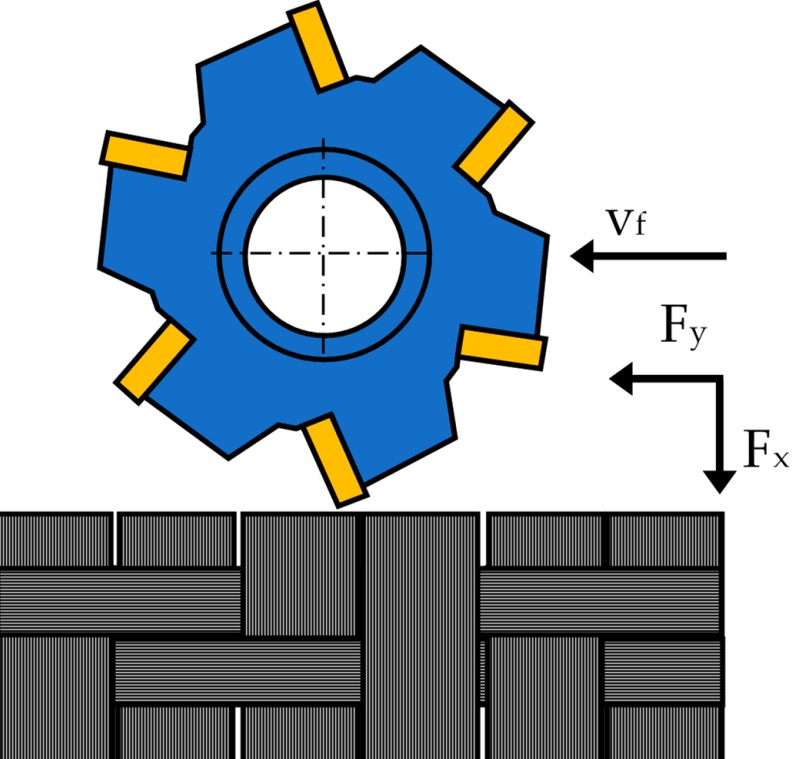

**Figure 5.** Orientation of the milling tool to the warp yarn.

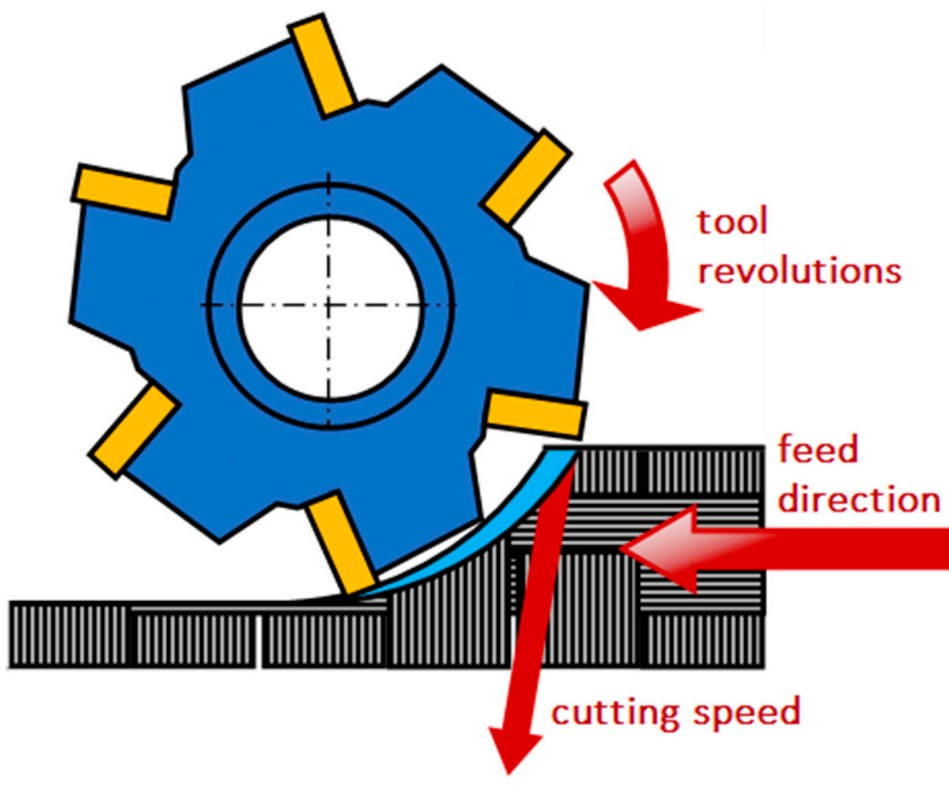

**Figure 6.** The arrangement of warp and weft yarns from the point of view of delamination during down milling.

### 3.1.1. G550 Milling Tool, UNICO Coating

Figure 7a shows the machined laminate surface with several protruding (approx. 0.5–2 mm) cut weft fibers. This is type II delamination. The protruding weft fibers are unevenly distributed over the entire machined surface. The machined surface does not have a clear cut, the edge is frayed. The image of tool Figure 7b indicates abrasive damage to the tool. The amount of wear is $187.60 \pm 0.20$ μm. The number of corrugations corresponds to the number of fabric layers in the laminate. Next, in Figure 7b, the yellow frame, abrasive damage to the tool is visible, due to protruding fibers.

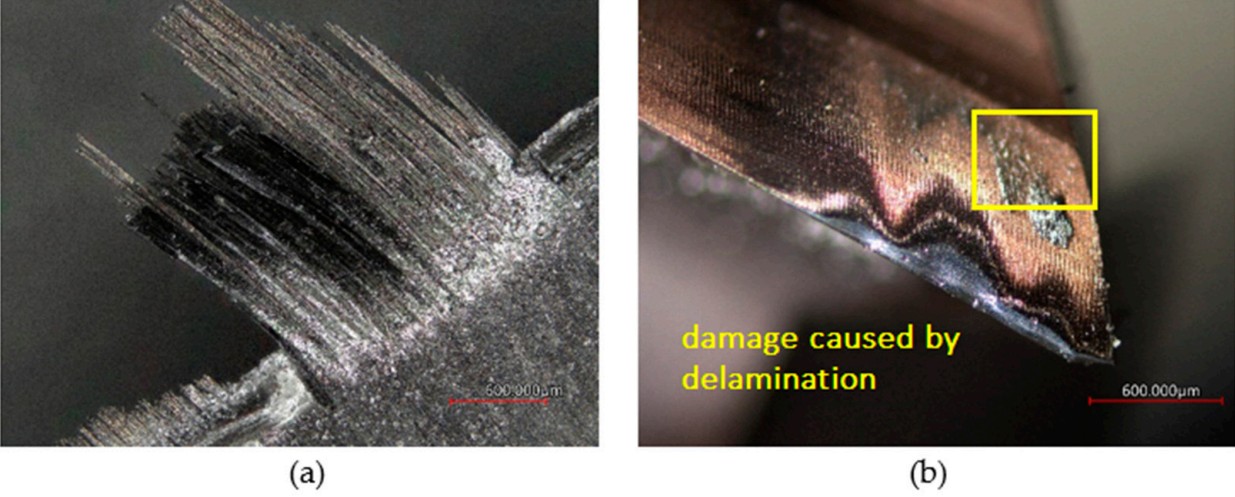

**Figure 7.** Capture after a tool tooth path of $15 \times 165$ m. (**a**) Laminate thickness 1 mm; (**b**) G550 milling tool.

Figure 8a shows a distinct layer of crushed resin on the fibers and many protruding (approx. 1.5 mm and more) cut weft fibers. Again, this is type II delamination. The protruding weft fibers are unevenly distributed over the entire machined surface. The machined surface does not have a clear cut, the edge is frayed. Furthermore, broken/separated bundles of warp yarn fibers also protrude from the machined surface. The image of tool Figure 8b shows significant abrasive damage to the tool. The size of the wear is 254.50 ± 0.28 μm (the shape of the wear can be compared to a sine function). The number of corrugations corresponds to the number of fabric layers in the laminate. Next, in Figure 8b, there is a yellow frame, where the abrasive damage of the tool due to the protruding fibers is again visible.

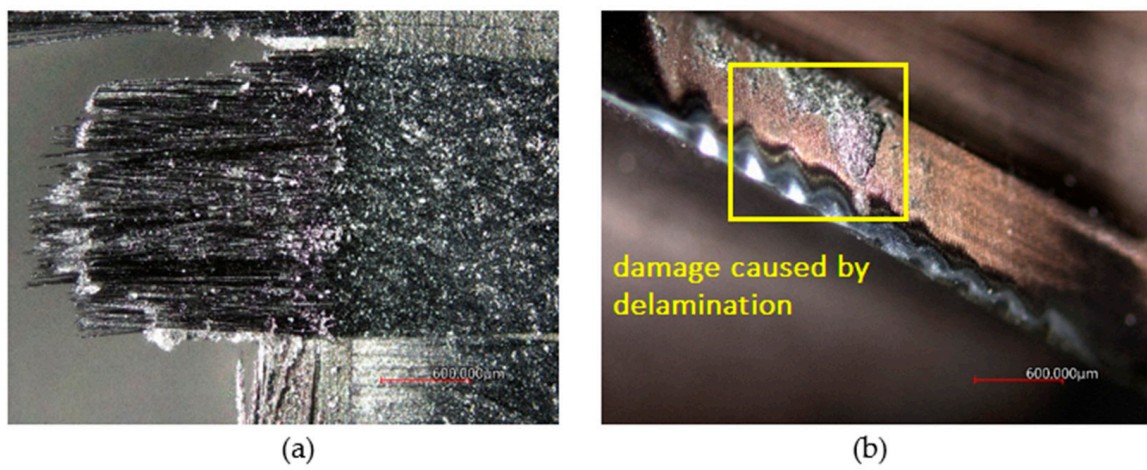

(a)  (b)

**Figure 8.** Capture after a tool tooth path of 15 × 165 m. (**a**) Laminate thickness 3 mm; (**b**) G550 milling tool.

### 3.1.2. A100 Milling Tool, CrN Coating

Figure 9a shows significant surface damage after milling with many protruding and pulled cut weft fibers (approx. 3 mm and more). Again, this is type II delamination. Protruding and broken weft fibers are unevenly distributed over the entire machined surface. The machined surface does not have a clear cut, the edge is significantly frayed to wavy and damaged. Broken/separated bundles of warp yarn fibers also protrude from the machined surface. Abrasive damage to the tool is visible in the image of tool Figure 9b. The amount of wear is 108.80 ± 0.20 μm. The number of corrugations corresponds to the number of fabric layers in the laminate.

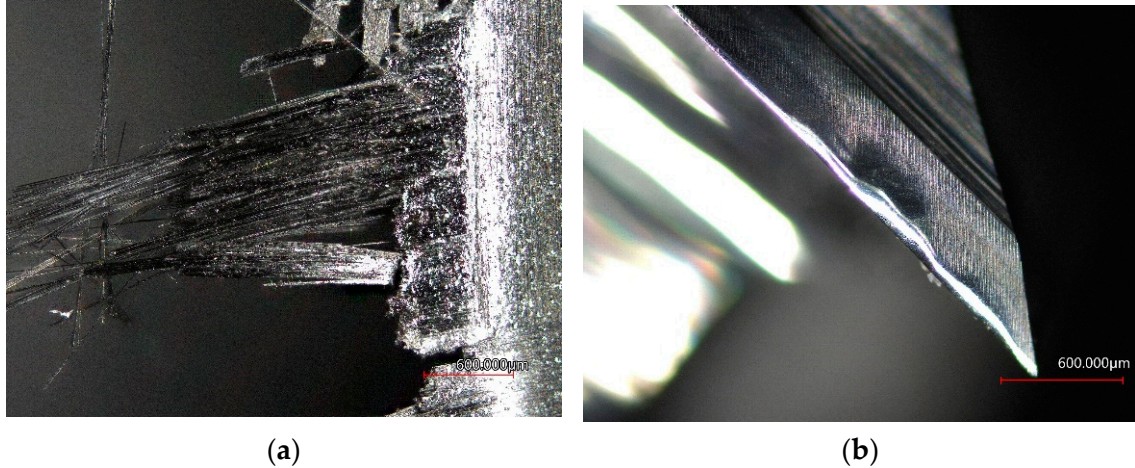

(**a**)  (**b**)

**Figure 9.** Capture after a tool tooth path of 15 × 165 m. (**a**) Laminate thickness 1 mm; (**b**) A100 milling tool.

Figure 10a shows a surface with many protruding fibers (approx. 3 mm and more) of cut weft fibers. Again, type II delamination. The protruding weft fibers are unevenly distributed over the entire machined surface. The machined surface does not have a clear cut, the edge is frayed, and slightly bent bundles of warp yarn fibers protrude from the machined surface. The image of tool Figure 10b shows abrasive damage to the tool. The amount of wear is 182.40 ± 0.20 μm. The number of corrugations corresponds to the number of fabric layers in the laminate.

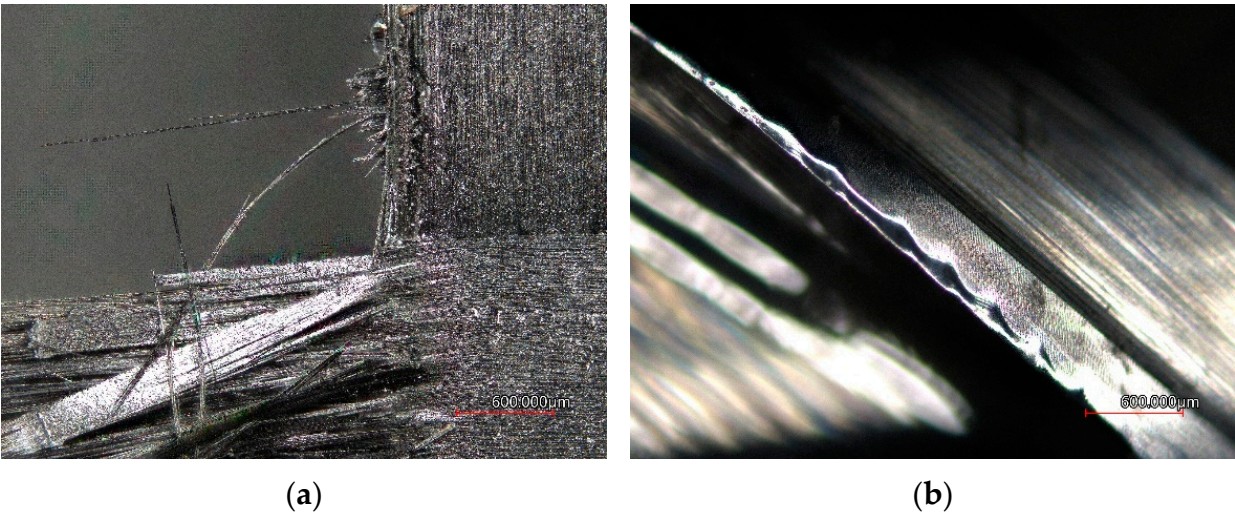

(**a**)           (**b**)

**Figure 10.** Capture after a tool tooth path of 15 × 165 m. (**a**) Laminate thickness 3 mm; (**b**) A100 milling tool.

### 3.1.3. ECSSF Milling Tool, DLC Coating

In Figure 11a, a very fine layer of destroyed (broken) resin on the laminate is observed. The machined surface has a clear cut, and the edge is solid, without burrs and protruding fibers. In some places of the machined surface, slightly protruding bent bundles of warp yarn fibers were rarely seen. In Figure 11b, the measured value of the tool coating wear can be observed, which is 284.550 μm due to the abrasive effects of the carbon fiber. Figure 11b of the tool indicates abrasive damage to the tool. The resulting wear rate of the tool base material was 74.70 ± 0.16 μm after the tool tooth path 15 × 165 m.

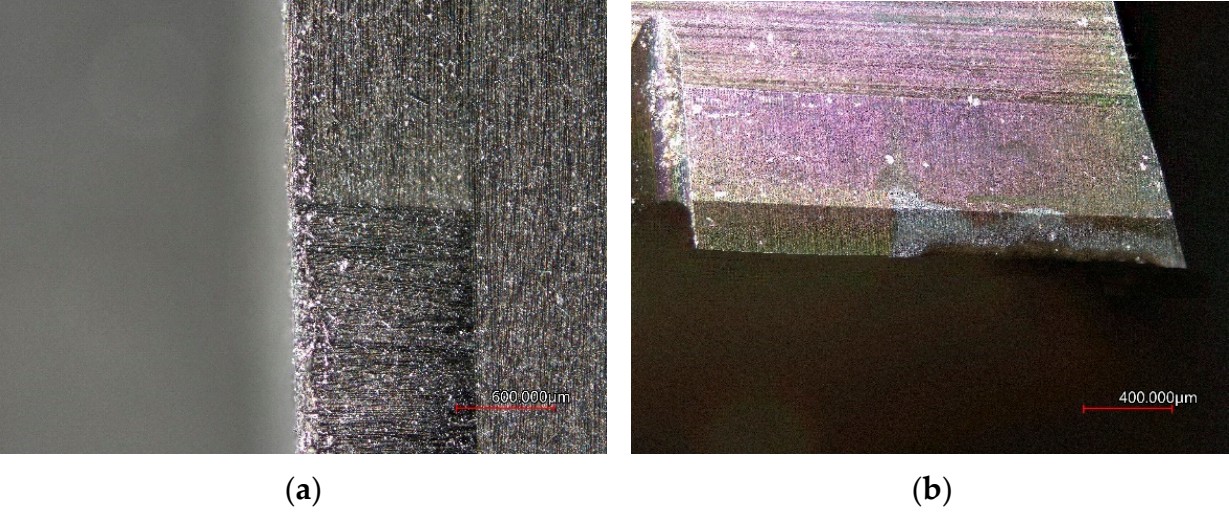

(**a**)           (**b**)

**Figure 11.** Capture after a tool tooth path of 15 × 165 m. (**a**) Laminate thickness 1 mm; (**b**) ECSSF milling tool.

In Figure 12a, a slight layer of destroyed (broken) resin on the fibers and a finely separated warp yarn bundle can be seen. The weft fibers are cut evenly, without protruding fibers. This is type I delamination, which occurs only in the plane of the first layer. The machined surface has a clear cut, but with slightly separated warp yarn bundles. The image of tool Figure 12b again points to the abrasive damage of the tool, the size of which is $98.40 \pm 0.18$ µm. The number of corrugations corresponds to the number of fabric layers in the laminate.

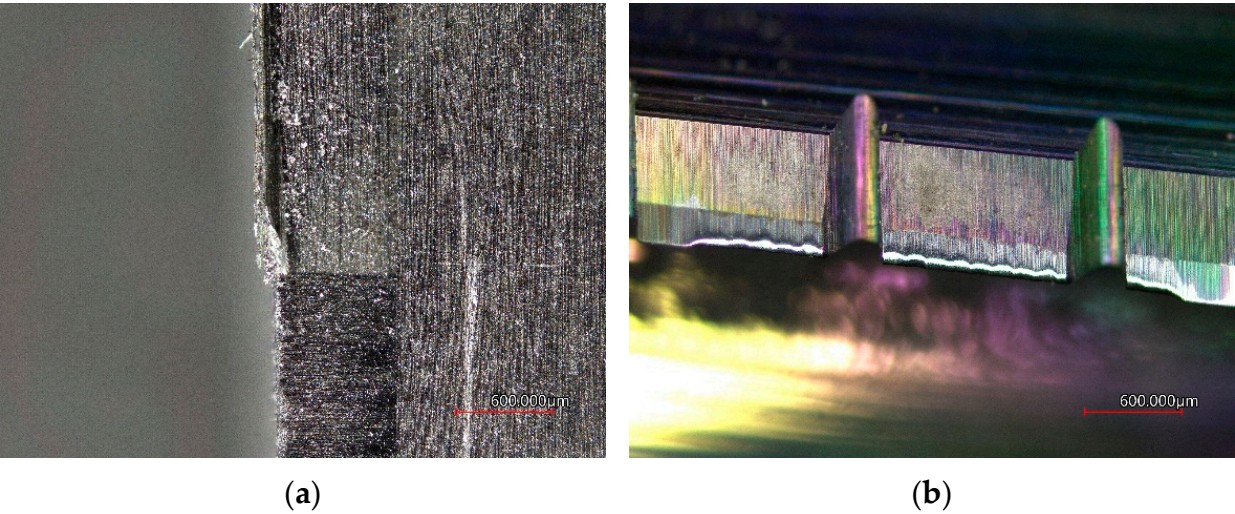

(**a**)               (**b**)

**Figure 12.** Capture after a tool tooth path of $15 \times 165$ m. (**a**) Laminate thickness 3 mm; (**b**) ECSSF milling tool.

*3.2. Cutting Forces Depending on Tool Wear*

The graphs below show the dependence of the recorded cutting forces on the amount of wear of the cutting tool. The individual reading intervals (after driving $5 \times 165$ m, $10 \times 165$ m, $15 \times 165$ m, and $20 \times 165$ m) of the measured values are marked with a point on the curve.

The highest measured values of the $F_x$ component of the cutting force at the beginning and at the end of the experiment were measured at the ECSSF stand, see Figure 13. The lowest measured value of the parameter $VB = 46.57 \pm 0.14$ µm corresponds to the size of the cutting force $F_x = 32.80 \pm 0.14$ N. These values were observed after traveling the tool path $5 \times 165$ m when machining a plate with a thickness of 1 mm.

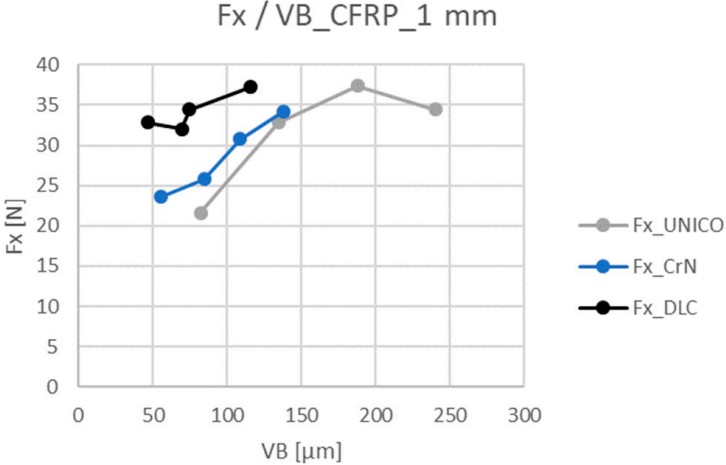

**Figure 13.** Dependence of the cutting force component $F_x$ on wear, laminate thickness 1 mm. $F_x$—cutting force in x direction. VB—flank wear.

The size of the VB wear produced on the A100 tool (137.16 ± 0.20 μm) is similar to the VB wear size of the DLC-coated tool (115.43 ± 0.20 μm). However, the course and measured values differ significantly. The A100 tool can be characterized by a linear development of the increase in the applied force $F_x$. A similar trend can be observed with the UNICO-coated tool.

Figure 13 shows the highest achieved tool wear of 240.05 ± 0.28 μm at the applied force $F_x$ = 34.40 ± 0.14 N. This value was measured with the G550 tool after traveling the tool path 20 × 165 m and is 1.75–2.1× greater than the other tools used, while the force $F_x$ achieves similar values. When using the tool with UNICO coating, the lowest force value $F_x$ = 21.60 ± 0.14 N was measured.

When machining a sample with a thickness of 3 mm according to the conditions, see Table 4, the results were measured and processed in Figure 14. The graph shows a similarity with Figure 13. Tools A100 and G550 show a similar course of applied forces, the ECSSF tool is characterized by the smallest measured amount of wear and the largest measured value of the applied force $F_x$. For both measured parameters, there was an approximately twofold increase in the measured values in the individual reading intervals for the A100 and G550 tools.

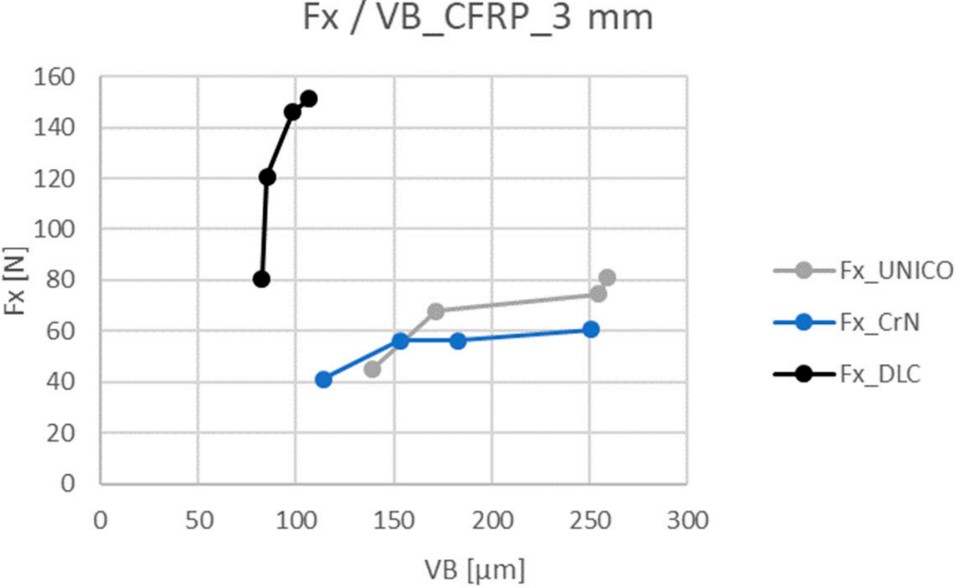

**Figure 14.** Dependence of the cutting force component $F_x$ on wear, laminate thickness 3 mm. $F_x$—cutting force in x direction. VB—flank wear.

The exception is the ECSSF tool. The size of the tool wear remained relatively the same, a significant change is an increase in the force $F_x$, where a 2.4–4 times increase in values can be observed.

From Figures 13 and 14, similarities in the course can be observed for tools with CrN and UNICO coating due to the similarity of tool geometries, see. Table 2.

The $F_y$ component of the cutting force was included among the evaluated parameters. From the following graphs, it is possible to observe an increase in the applied forces compared to the $F_x$ component by 10 to 30 N for the A100 and G550 tools for a sample with a thickness of 1 mm, see Figure 15. For a sample with a thickness of 3 mm, see Figure 16, there is an increase in the magnitude of the forces by 20 to 50 N.

The ECSSF tool is characterized by lower $F_y$ forces by 10 to 40 N compared to the $F_x$ component. The size of the $F_y$ component is significantly influenced by the geometry of the tool used and the size of the tool feed.

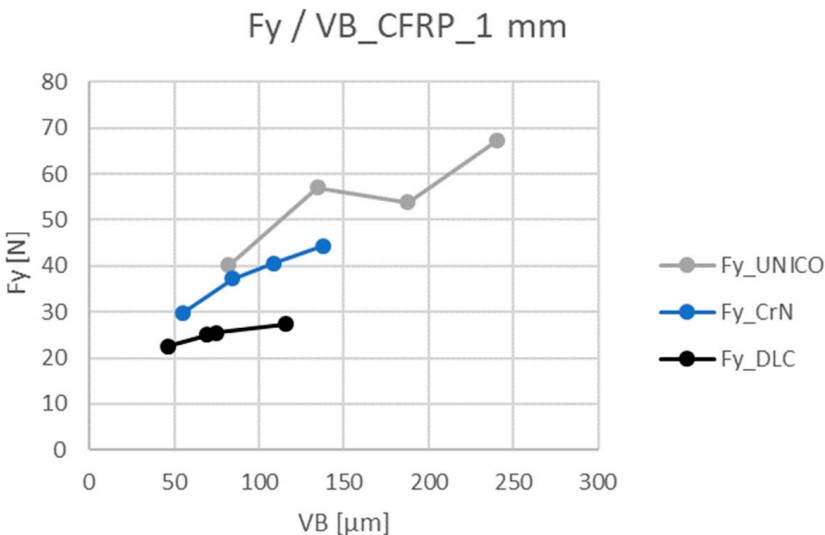

**Figure 15.** Dependence of the cutting force component $F_y$ on wear, laminate thickness 1 mm. $F_x$—cutting force in x direction. VB—flank wear.

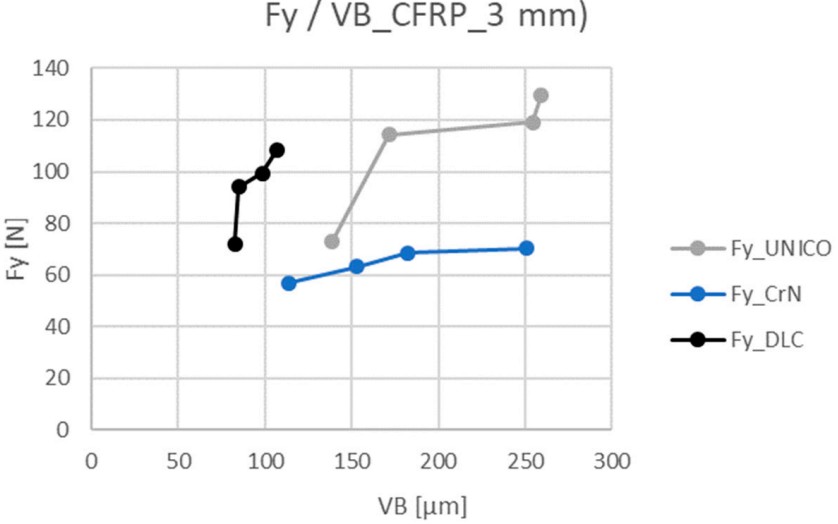

**Figure 16.** Dependence of the cutting force component $F_y$ on wear, laminate thickness 3 mm. $F_x$—cutting force in x direction. VB—flank wear.

From Figure 15, it is possible to observe an increase in the acting forces with increasing helix pitch angle. The individual curves of the dependences of the acting component of the cutting force $F_y$ depending on VB during the machining of a sample with a thickness of 3 mm show a similar course as that of the component $F_x$ for the same machined thickness.

The curve for the tool with the DLC coating again shows a steep increase in the applied force $F_y$, but the values of forces $F_x$ in the order of 10 N are not reached. A significant difference between the measured components is visible for the UNICO tool. The CrN tool achieves similar values for both measured force components.

### 3.3. Roughness Parameters of the Machined Surface Depending on Tool Wear

The parameters $R_a$, $R_z$ and $R_t$ were determined as roughness evaluation parameters due to their frequent use in the technical practice of European countries. According to Figure 17, an increase in the values of the parameter $R_a$ can be observed for all the curves with increasing wear of the VB. Figures 11a and 17 show the achieved surface quality when using the ECSSF tool supported by the measured values. The difference between the measured values at the beginning and end of the measurement is 1.16 μm. After driving

the 20 × 165 m track, the G550 tool shows an increase in the value of the $R_a$ parameter by 3.02 µm and the A100 tool an increase by 2.55 µm. A more pronounced increase in the values of the $R_a$ parameters was measured for the A100 and G550 tools. The G550 tool shows an asymptotic growth to a value of 6 µm after traveling a tool path of 20 × 165 m. The A100 tool shows tendencies towards a further increase in the values of the $R_a$ parameter.

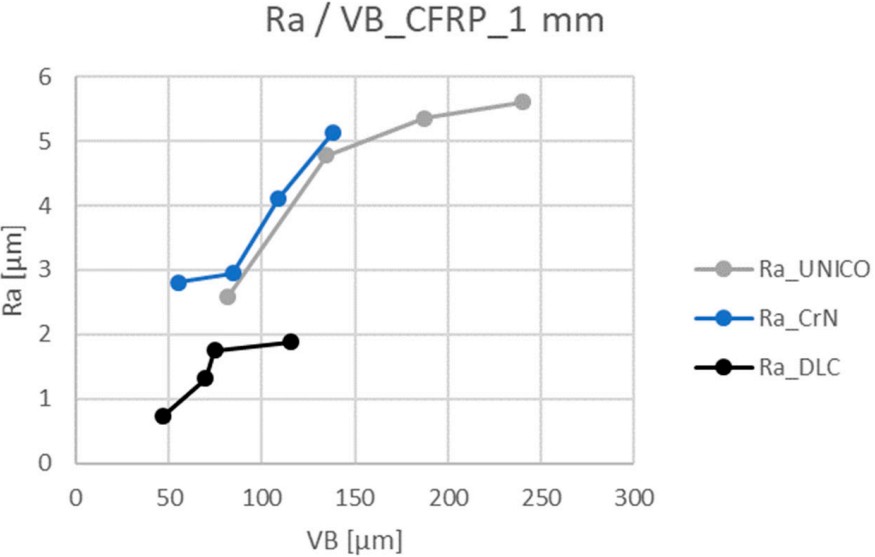

**Figure 17.** Dependence of the surface roughness parameter $R_a$ on wear, laminate thickness 1 mm. $R_a$—arithmetical mean high. VB—flank wear.

From the measured values when machining a sample with a thickness of 3 mm, see Figure 18, a significant change in the course of the individual graphs can again be observed. The most significant change is seen before VB = 150 µm. This is where the graphs for the A100 and G550 intersect. The G550 tool shows a steep increase in the values of the parameter $R_a$ to the value $R_a$ = 6 µm at VB = 171.56 ± 0.20 µm. The course of the graph for the A100 tool shows a smoother course and asymptotically approaches the value of $R_a$ = 4 µm.

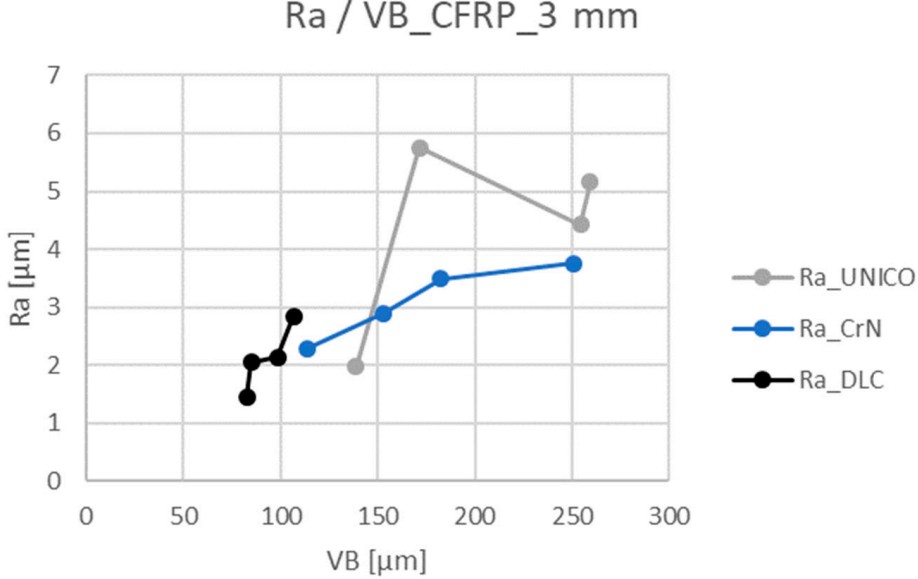

**Figure 18.** Dependence of the surface roughness parameter $R_a$ on wear, laminate thickness 3 mm. $R_a$—arithmetical mean high. VB—flank wear.

Other relevant parameters for evaluating the quality of the surface after machining are the roughness parameters $R_z$ and $R_t$. These parameters make it possible to describe in more detail the structure of the surface resulting from the occurrence of delamination, non-cutting of fibers, crumbling of the matrix, etc. Graphs showing the dependence of parameters $R_z$ and $R_t$ on the amount of VB wear show significant similarity to Figures 17 and 18.

When comparing Figures 19 and 20, a decrease in the measured values of the $R_z$ parameter can be observed as the wear of the VB tool increases and the machined thickness of the sample increases. A decrease can be observed in the A100 and G550 tools. With the ECSSF instrument, the increase in measured values is noticeable mainly by 50%.

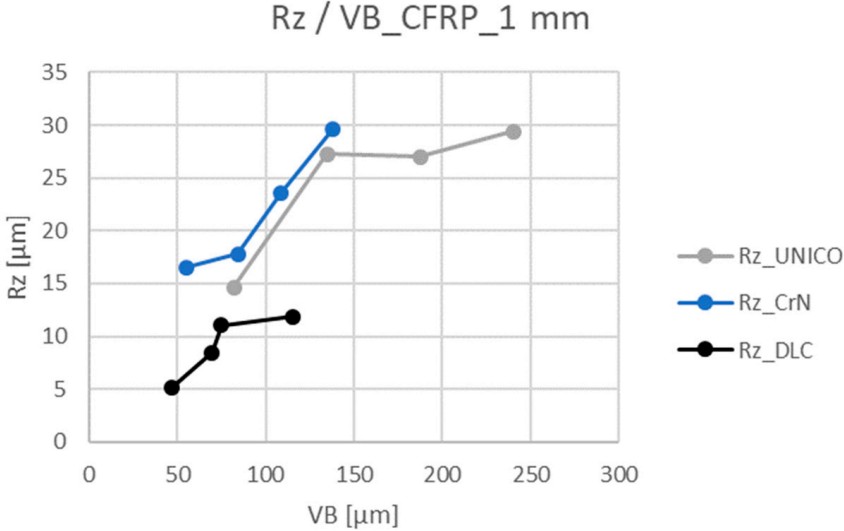

**Figure 19.** Dependence of the surface roughness parameter $R_z$ on wear, laminate thickness 1 mm. $R_z$—maximum height of profile. VB—flank wear.

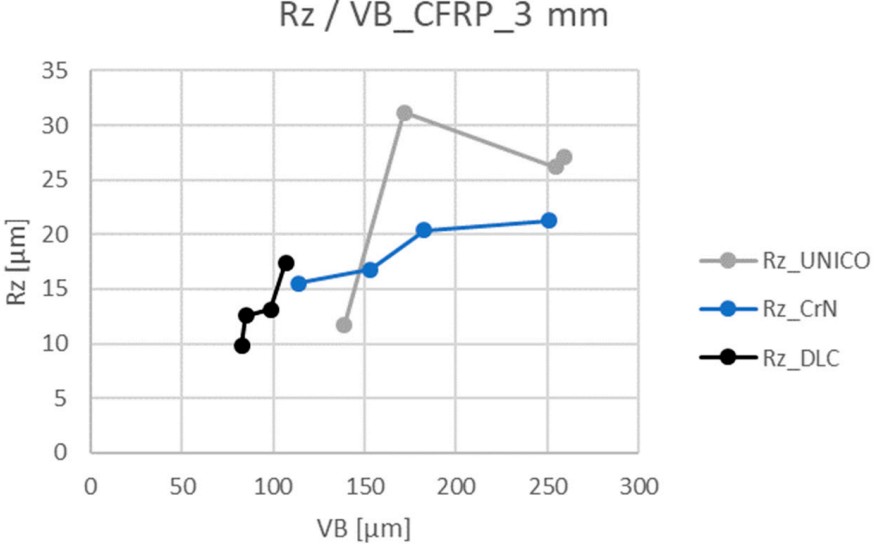

**Figure 20.** Dependence of the surface roughness parameter $R_z$ on wear, laminate thickness 3 mm. $R_z$—maximum height of profile. VB—flank wear.

The depth of roughness $R_t$, which is the sum of the highest peak of the profile and the depth of the deepest depression of the R profile inside the measured path for a plate thickness of 1 mm, reaches up to $10\times$ greater values compared to the $R_a$ parameter, see Figures 17 and 19. Another noticeable increase in the measured values is compared to the $R_z$ parameter, Figure 21. Measured the values for the tool with DLC coating show an

increase in values of about 5 μm, for tools with CrN and UNICO coatings the increase is about up to 20 μm.

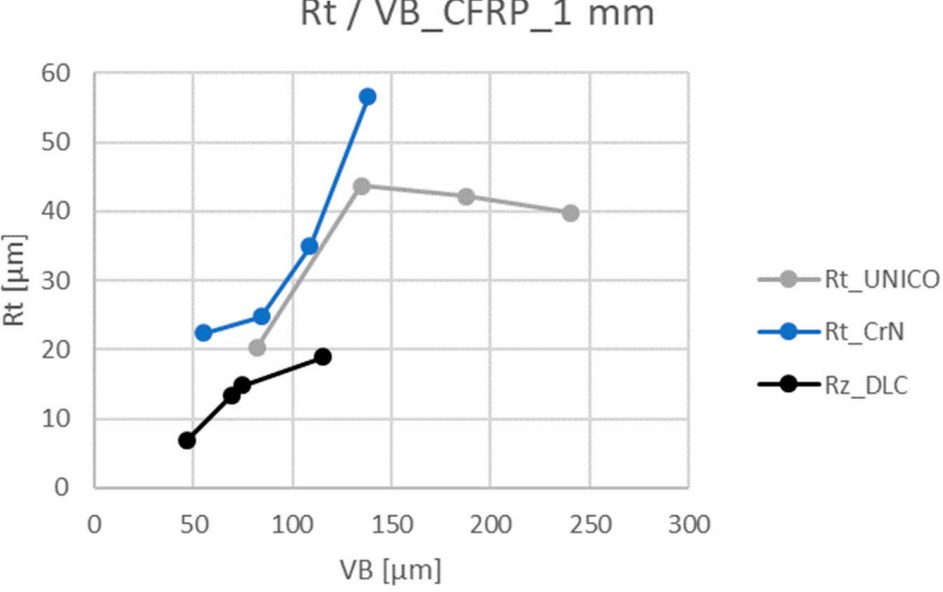

**Figure 21.** Dependence of the surface roughness parameter $R_t$ on wear, laminate thickness 1 mm. $R_t$—total height of profile. VB—flank wear.

A similar course of the increase in values is also visible in a sample with a thickness of 3 mm, Figure 22. Here, individual tools show a difference in $R_t$ parameter values of 4–10 μm compared to the $R_z$ parameter, see Figures 20 and 22.

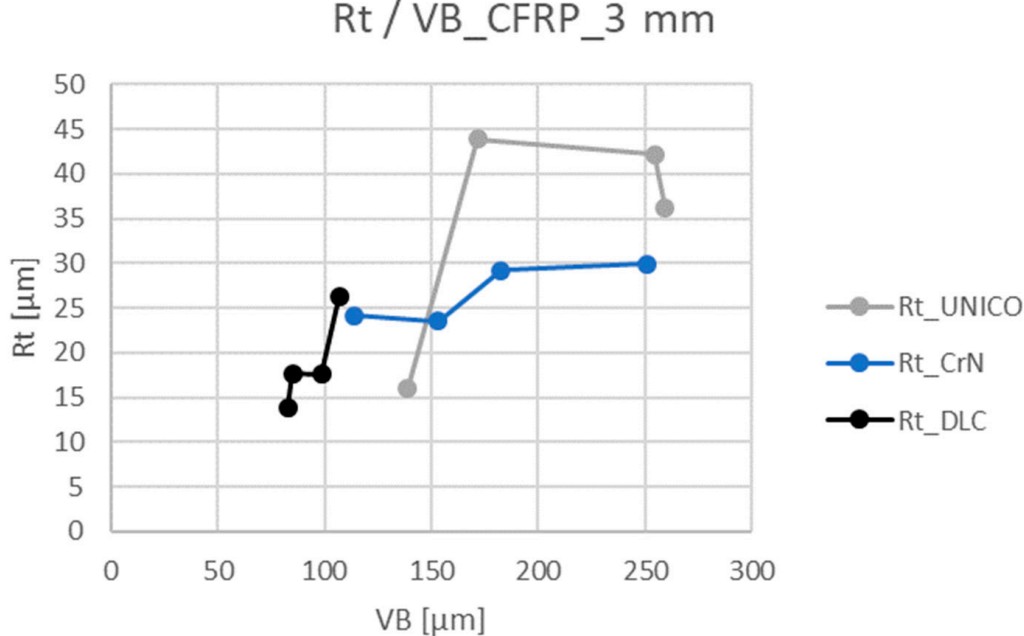

**Figure 22.** Dependence of the surface roughness parameter $R_t$ on wear, laminate thickness 3 mm. $R_t$—total height of profile. VB—flank wear.

### 3.4. Overall Comparison of the Resulting Wear of Individual Tools

Across the experiments, after evaluating the VB tool wear parameter over a tool path of 5 × 165 m, 10 × 165 m, 15 × 165 m, and 20 × 165 m, when machining 1 mm and 3 mm thick samples, the highest wear values were achieved with the UNICO coated tool in all controlled sections, see Figures 23–30.

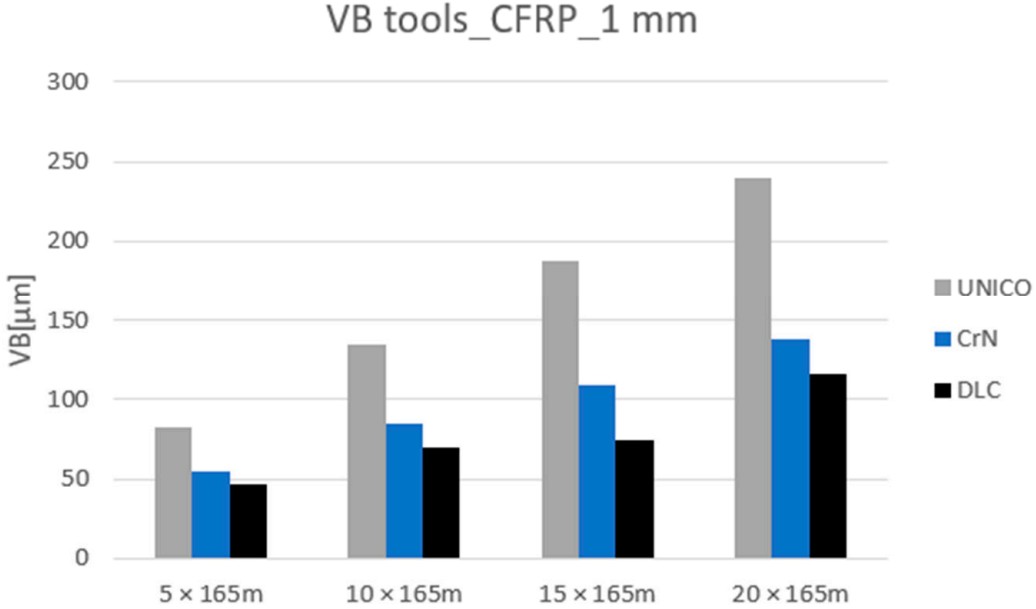

**Figure 23.** Tool wear, laminate thickness 1 mm.

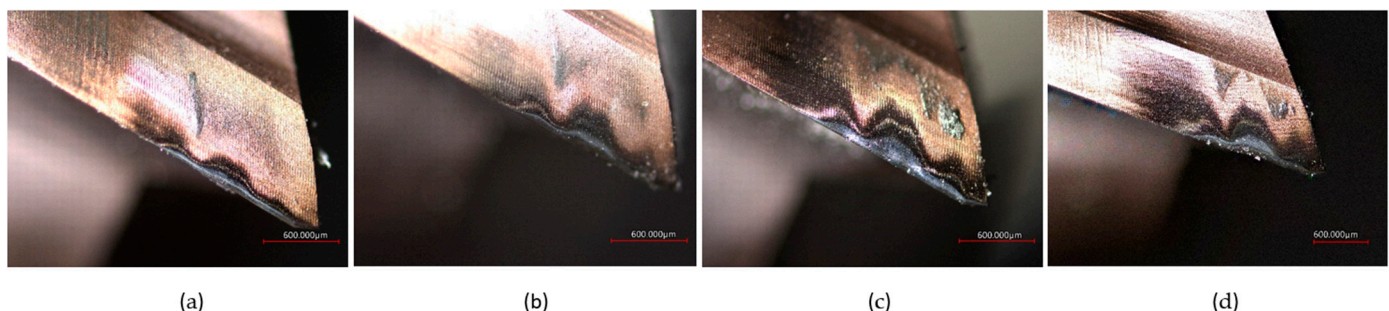

| (a) | (b) | (c) | (d) |

**Figure 24.** Tool wear of the G550 milling tool, laminate thickness 1 mm. (**a**) Tool tooth path 5 × 165 m; (**b**) tool tooth path 10 × 165 m; (**c**) tool tooth path 15 × 165 m; (**d**) tool tooth path 20 × 165 m.

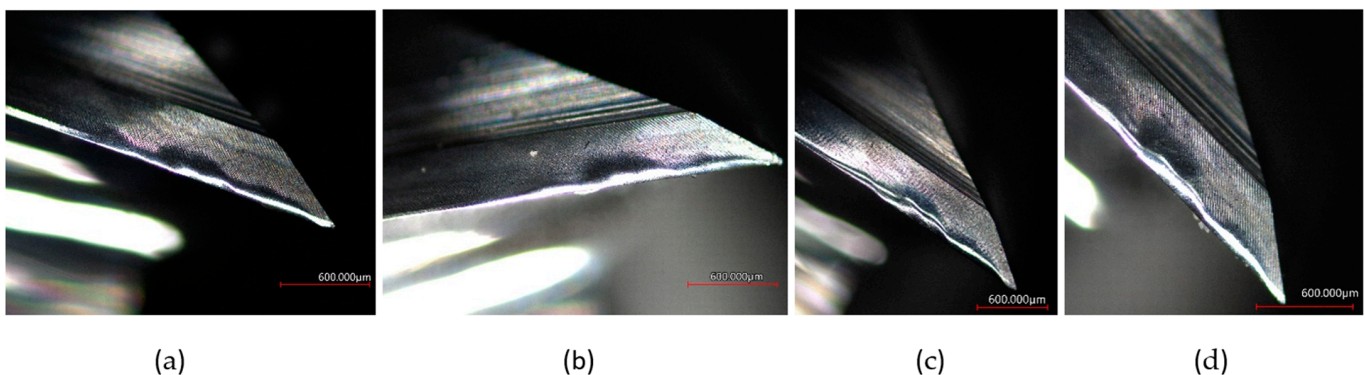

| (a) | (b) | (c) | (d) |

**Figure 25.** Tool wear of the A100 milling tool, laminate thickness 1 mm. (**a**) Tool tooth path 5 × 165 m; (**b**) tool tooth path 10 × 165 m; (**c**) tool tooth path 15 × 165 m; (**d**) tool tooth path 20 × 165 m.

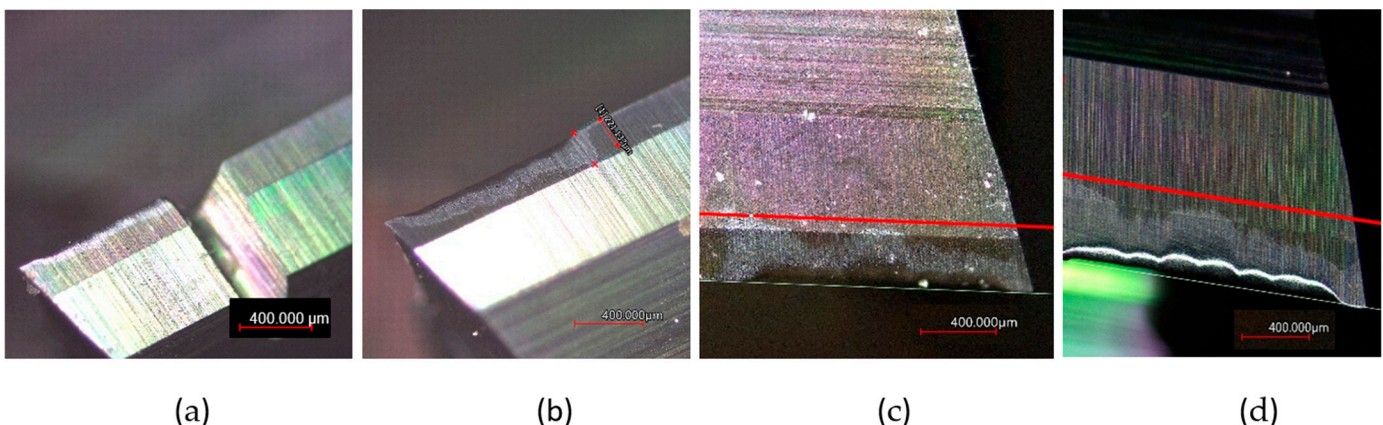

**Figure 26.** Tool wear of the ECSSF milling tool, laminate thickness 1 mm. (**a**) Tool tooth path 5 × 165 m; (**b**) tool tooth path 10 × 165 m; (**c**) tool tooth path 15 × 165 m; (**d**) tool tooth path 20 × 165 m.

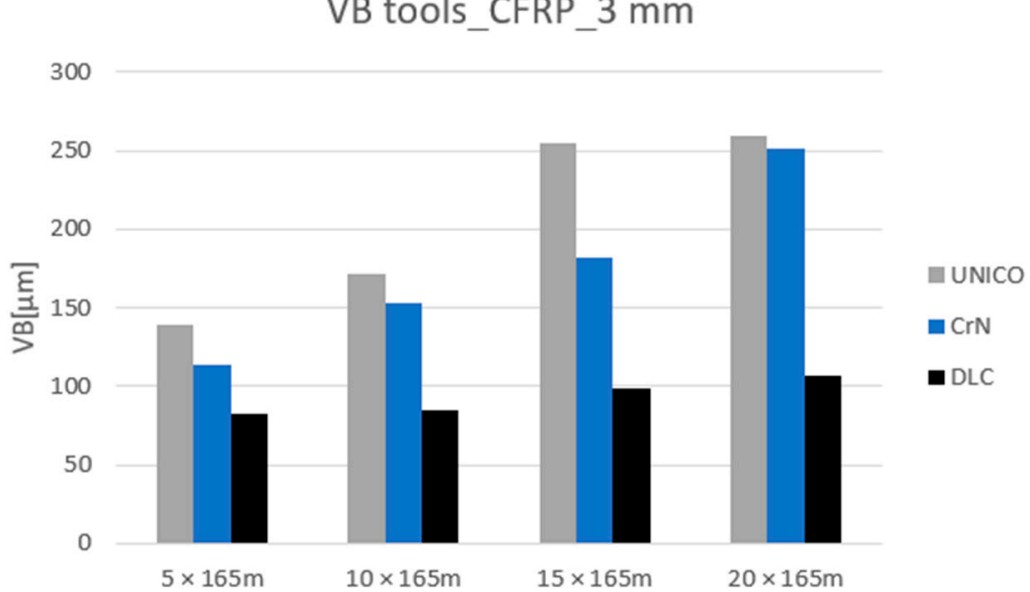

**Figure 27.** Tool wear, laminate thickness 3 mm.

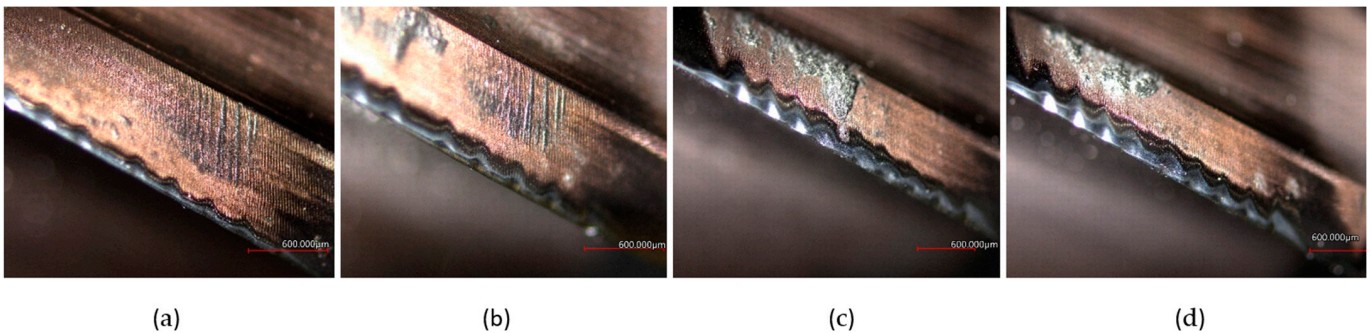

**Figure 28.** Tool wear of the G550 milling tool, laminate thickness 3 mm. (**a**) Tool tooth path 5 × 165 m; (**b**) tool tooth path 10 × 165 m; (**c**) tool tooth path 15 × 165 m; (**d**) tool tooth path 20 × 165 m.

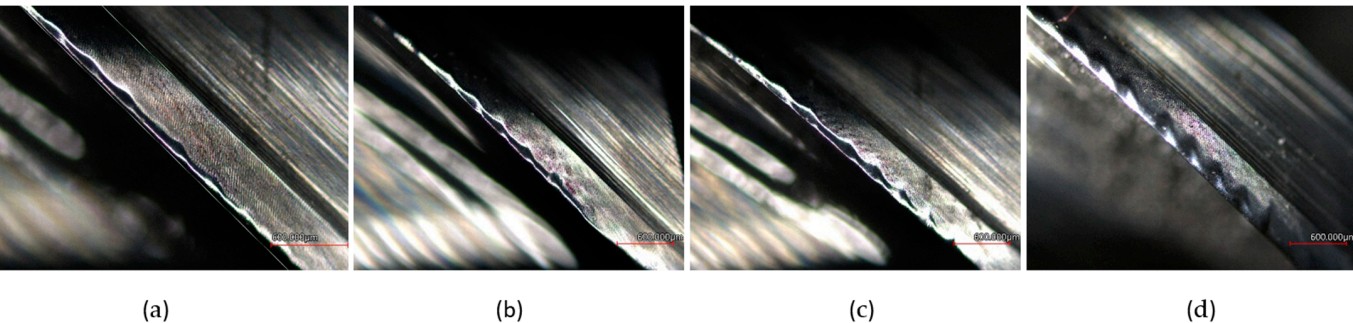

|  (a)  |  (b)  |  (c)  |  (d)  |

**Figure 29.** Tool wear of the A100 milling tool, laminate thickness 3 mm. (**a**) Tool tooth path 5 × 165 m; (**b**) tool tooth path 10 × 165 m; (**c**) tool tooth path 15 × 165 m; (**d**) tool tooth path 20 × 165 m.

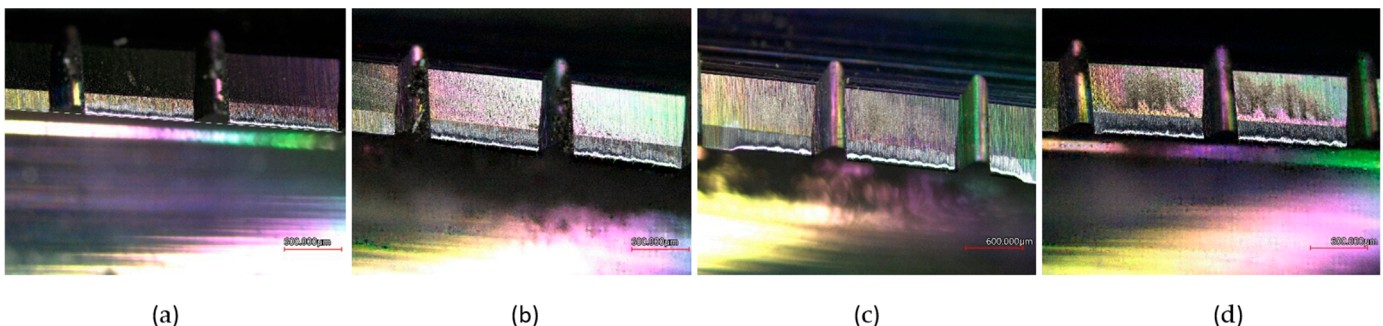

|  (a)  |  (b)  |  (c)  |  (d)  |

**Figure 30.** Tool wear of the ECSSF milling tool, laminate thickness 1 mm. (**a**) Tool tooth path 5 × 165 m; (**b**) tool tooth path 10 × 165 m; (**c**) tool tooth path 15 × 165 m; (**d**) tool tooth path 20 × 165 m.

This tool was the only tool used to show wear in the form of nicks and scratches extending into the base material of the tool outside the cutting area, see Figures 7b and 8b. This phenomenon was caused by the abrasive effects of uncut fibers and the influence of the helix rising angle of the cutter, which was the largest of all geometries used. The high rate of tool wear had a significant effect on the monitored roughness parameters. In most cases, the examined roughness parameters reached the highest values in the control sections (5×, 10×, 15×, and 20 × 165 m) with the G550 tool. The length of the protruding weft fibers reached slightly lower values compared to the CrN tool, in the order of 0.5–3 mm.

The A100 tool shows the second highest measured wear values. Similar magnitudes of values can be observed for other examined parameters of forces and roughness. During the machining of the sample with a thickness of 3 mm, the magnitudes of the resulting wear are very similar to the values of the G550 tool due to the similarity of the geometry, which was manifested in a significant way at this thickness.

The lowest VB values were achieved with the ECSSF tool, which excelled in wear resistance. Furthermore, the lowest values of the surface roughness parameters and the highest quality of the machined surface were achieved among all the examined samples. The machined edge shows a minimal amount of protruding warp fibers and burrs. The application of this solution eliminated the appearance of protruding weft fibers, fiber breakage, and the formation of burrs in the form of warp fibers.

## 4. Discussion

After machining the material with tools A100 and G550, the material shows damage to the surface by delamination. The resulting delamination also results in the wear of the tool outside the cutting site. The surfaces of all samples machined with the mentioned tools showed protruding weft fibers from 0.5–3 mm or more, or separate bundles of warp yarn fibers. Another phenomenon was crushed resin or carbon fiber particles on the surfaces

of the samples (the geometry of the cutters prevented the removal of resin and carbon fiber particles).

According to [19], the max. length of protruding fibers in the yarn length direction corresponded to the distance between the initial contact of the cutting edge with the fiber and the resulting milled edge. Further, the claim of the authors [16] was confirmed that fracture energy of the matrix/fiber interface is constant along the undulating yarn and the support by the resin layer varies in a sinusoidal way. A crack can only propagate further as long as the energy induced by bending, which degrades along the crack propagation, exceeds the necessary fracture energy of the resin layer and the matrix/fiber interface.

Conversely, the laminate machined with the ECSSF cutter and DLC coating showed no damage to the machined surface. The machined surface had a clear cut, without significant pulled out or protruding weft or warp fibers.

The graphs showing the dependence of cutting forces $F_x$ and $F_y$ on the amount of wear of the cutting tool show the expected increase in force values with the increasing amount of wear.

With the ECSSF tool, a sharp increase in forces can be observed in the graphs processing the $F_x$ component when machining a sample thickness of 3 mm, while maintaining similar final tool wear values of $115.43 \pm 0.20$ µm for a thickness of 1 mm and $106.82 \pm 0.20$ µm for a thickness of 3 mm. There was a significant increase in the acting forces already when measuring after traveling the tool path $5 \times 165$ m. A thickness of 1 mm showed a force $F_x = 32.80 \pm 0.14$ N during wear $VB = 46.57 \pm 0.14$ µm, a thickness of 3 mm $VB = 82.56 \pm 0.16$ µm a force $F_x = 80.60 \pm 0.16$ N. In other control sections, the individual measured forces $F_x$ showed up to a four times greater increase with relatively similar values of wear VB. Cutting forces $F_x$ and $F_y$ on the amount of wear of the cutting tool show the expected increase in the values of forces with the increasing amount of wear.

From the processed results, it is possible to observe the mutual similarity of the courses of the investigated quantities for the A100 and G550 tools due to the similarity of the geometry. For these tools, a significant increase in the size of the final wear of the VB and the components of the cutting force was observed. The A100 tool showed wear and force components when machining a thickness of 1 mm $VB = 138.16 \pm 0.20$ µm, $F_x = 34.20 \pm 0.14$ N and $F_y = 44.40 \pm 0.14$ N, when machining a thickness of 3 mm $VB = 250.73 \pm 0.28$ µm, $F_x = 60.60 \pm 0.16$ N and $F_x = 70.20 \pm 0.16$ N. Similarly, the G550 tool $VB = 240.05 \pm 0.28$ µm, $F_x = 34.40 \pm 0.14$ N, $F_y = 67.20 \pm 0.16$ N for a machined thickness of 1 mm and when machining a thickness of 3 mm $VB = 259.21 \pm 0.28$ µm, $F_x = 81.20 \pm 0.16$ N and $F_y = 129.60 \pm 0.20$ N.

The geometry of the cutting edge and the orientation of the fibers at contact play a critical role in tool wear and the final roughness of the machined surfaces. Tool flank face wears affected the machined surface structure, roughness, and topography. The more the tool was worn, the more the laminate surface was devastated. When the weft fibers were oriented perpendicular to the tool cut, the A100, CrN coated and G550, UNICO coated cutters caused a strong bending of the weft fibers, which resulted in a poor-quality cut of the fibers, which was manifested in the form of ragged protruding fibers, Figures 8a and 9a. There was serious damage to the surface with the carbon fibers being crushed or chopped and pulled out. The opposite was the case with the ECSSF cutter, and DLC coating, Figures 11a and 12a.

The accompanying phenomena mentioned above significantly influenced the final quality of the machined surfaces. The parameters $R_a$, $R_z$, and $R_t$ were chosen as the surface quality evaluation parameters, which for these two tools reached significantly higher values when machining samples with a thickness of 1 and 3 mm than the ECSSF tool.

Low surface roughness and low delamination characterize the final quality of machined surfaces when using the ECSSF tool, DLC coating. On the machined surface topographies, the ECSSF cutter, and DLC coating showed surfaces with smooth surface textures, despite the effect of vibrations.

Measured values after the tool path $20 \times 165$ m:

- G550 tool, UNICO coating, 1 mm thickness-$R_a$ = 5.60 $\pm$ 0.10 μm, $R_z$ = 29.40 $\pm$ 0.14 μm, $R_t$ = 39.77 $\pm$ 0.14 μm;
- G550 tool, UNICO coating, 3 mm thickness-$R_a$ = 5.16 $\pm$ 0.10 μm, $R_z$ = 27.13 $\pm$ 0.14 μm, $R_t$ = 36.19 $\pm$ 0.14 μm;
- A100 tool, CrN coating, 1 mm thickness-$R_a$ = 5.13 $\pm$ 0.10 μm, $R_z$ = 29.65 $\pm$ 0.14 μm, $R_t$ = 56.59 $\pm$ 0.14 μm;
- A100 tool, CrN coating, thickness 3 mm-$R_a$ = 3.75 $\pm$ 0.10 μm, $R_z$ = 21.28 $\pm$ 0.14 μm, $R_t$ = 29.95 $\pm$ 0.14 μm;
- ECSSF tool, DLC coating, 1 mm thickness-$R_a$ = 1.89 $\pm$ 0.10 μm, $R_z$ = 11.86 $\pm$ 0.12 μm, $R_t$ = 18.93 $\pm$ 0.14 μm;
- ECSSF tool, DLC coating, 3 mm thickness-$R_a$ = 2.85 $\pm$ 0.10 μm, $R_z$ = 17.42 $\pm$ 0.14 μm, $R_t$ = 26.23 $\pm$ 0.14 μm.

According to Figures 13 and 14, there are noticeable differences in the amount of wear of individual tools at the end of individual control sections. The lowest measured wear size values were achieved across all parameters and control sections by the ECSSF tool.

Against the ECSSF tool, DLC coating, the rest of the tools will be compared, and the achieved measurement results will be compared. The G550 tool shows values higher on average by 107% (in individual sections 76, 93, 151, and 108%) when machining a thickness of 1 mm and 118% (in individual sections 68, 102, 159, and 143%) for a machined thickness of 3 mm. The A100 tool achieves higher values when machining a sample with a thickness of 1 mm, by an average of 26% (18, 21, 46, and 20% in individual sections).

When machining a sample with a thickness of 3 mm, higher values were achieved by an average of 85% (38, 80, 85, and 135% in individual sections). The achieved results are in accordance with various studies [19–24].

## 5. Conclusions

The main problem in the machining of CFRP laminate is the delamination of the layers (within the cut) and permanent tool damage due to inappropriate tool selection. This experimental study investigated the wear effect of selected types of cutters suitable (ECSSF cutter, DLC coating) or less suitable (A100 cutter, CrN, and G550 cutter, UNICO coating) for milling CFRP laminate with twill weave fabric.

Based on the study, the following conclusions were drawn:

1. Significant damage to the laminate/sample surface occurs with less suitable cutters (the A100 carbide cutter, which is primarily intended for machining non-ferrous metals, aluminum, and aluminum alloys, and the G550 universal cutter suitable for steel, stainless steel, cast iron, and hardened material)—which was also a prerequisite before starting the study.
2. The surface machined with the ECSSF cutter, DLC coating (designed for CFRP/GFRP laminate composite materials) had a clear cut with a smooth texture, despite the effects of vibration.
3. The surface machined with A100 cutters, CrN coating and G550, UNICO coating showed a poor fiber cut, manifested in the form of ragged protruding fibers and surface damage.
4. The increase in the amount of wear of individual tools significantly influenced the course of the forces $F_x$ and $F_y$ acting on the tool and the resulting measured roughness parameters. Due to the increase in wear, the smooth removal of individual bundles of fibers (warp and weft) was not ensured, the abrasive effect of pulled/uncut fibers on the tool increased and worn areas outside the cut surface were formed. These factors and the local inhomogeneity of the sample due to the manufacturing process resulted in a decrease in the values of the measured parameters in some control sections. The surface roughness parameters were significantly affected by the extent of delamination.
5. The CrN-coated A100 tool achieves relatively similar results of the monitored parameters to the DLC-coated ECSSF tool when machining a 1 mm thick sample.

6.  Research in this direction should continue to investigate other possible types of tool geometries and tool coatings in order to complete the research results and determine the optimal conditions for machining composite materials.

**Author Contributions:** Conceptualization, T.K. and Š.D.; methodology, Š.D.; validation, Š.D.; formal analysis, Š.D.; investigation, A.K. and T.K.; resources, A.K. and T.K.; data curation, A.K.; writing—original draft preparation, T.K.; writing—review and editing, T.K.; visualization, A.K.; supervision, Š.D. All authors have read and agreed to the published version of the manuscript.

**Funding:** Research was funded by institutional funding of science and research at the Technical University of Liberec and by the Student Grant Competition of the Technical University of Liberec under the project number SGS-2022-5043—"Research and development in the field of machining of metal and composite materials using new knowledge for industrial practice".

**Institutional Review Board Statement:** Not applicable.

**Informed Consent Statement:** Not applicable.

**Data Availability Statement:** Data sharing is not applicable to this article.

**Conflicts of Interest:** The authors declare no conflict of interest. The funders had no role in the design of the study; in the collection, analyses, or interpretation of data; in the writing of the manuscript; or in the decision to publish the results.

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
