# Peer review of "Wear Study of Coated Mills during Circumferential Milling of Carbon Fiber-Reinforced Composites and Their Influence on the Sustainable Quality of the Machined Surface"

_coatings, doi:10.3390/coatings12101379_

Round 1
Reviewer 1 Report
This paper reports an investigation on the wear study of coated mills during milling of CFRP and its influence on the quality of machined surface. After reviewing this technical paper several times the reviewer believes the current form has a distance before accept in coatings. Some comments are given as follows.
Major comments:
The authors used the tools with the same diameter, but with different geometry, different coating, different number of teeth. How to define the effect of every single factor on the wear and cutting responses? This should be explained in the paper.
Only one case of cutting condition was used. So, the conclusion has been limited. The authors should add the results with different conditions to make the paper more scientific.
The authors show some pictures of the edge and tool wear with different laminate thickness. However, the evolution of the results was not shown. This is not rigorous for the wear study and its effect on the surface quality. The authors should add the evolution of the results with different wear stages.
The analysis for the cutting force evolution with different tool wear is not scientific. The author should make some explanation from the perspective of friction between the tool and the workpiece.
Minor comments:
The errors bars for some results are missing. For example, Graph 01-10 and chart 01-02.
The number of the figures are not consistent. Please check.
The English should be improved since some grammar error was noticed.
Reviewer 2 Report
Manuscript ID: coatings-1863908
Manuscript title: Wear study of coated mills during circumferential milling of carbon fiber reinforced composites and their influence on the sustainable quality of the machined surface
In my opinion, the manuscript should be substantially corrected as in its present form it lacks scientific merit. In the present form, it fully resembles an industrial report.
The major issues to be addressed are as follows:
- There is no clear objective of the manuscript/research set at the end of Introduction as it mandatory should be the case. As meatal cutting/cutting tool development is an application science, a clear technical objective of the study should be formulated and then, in conclusions, the benefits of the results of the study for theory and applications in comparison with the existing results should be presented. This is not the case in the manuscript.
- Work material: Only very generic description of the work materials used are provided with no properties or at least make. Note that a fiber-reinforced plastic (CFRP) of the same structure but made by different material manufacturers may have considerably different properties which significantly affect its machinability.
- Tool materials: There is not particular grades of the carbide of the endmills used in the tetes are indicated. Instead, only the mysterious harnesses of the coating layers are shown. Neither the method of measurement of this hardness nor the thickness of the coatings are indicated. Note that the tool wear patter shown in the manuscript (e.g. in Figure 06) clearly indicate that the tool worn much beyond the coating thickness so that the properties of the sintered carbide it is made of is of prime importance.
- Tool geometry: there is no parameters of the cutting tool geometry indicated for example, clearance and rake angles, flute profile, etc.
- Tool microgeometry. The most important parameter as the radius of the cutting edge is not indicated for the tools used. Note, because the feed per tooth is 20 microns, this parameter of crustal importance for tool performance including the cutting force, roughness of the machined surface including crushed resin, fiber pull-offs, etc. No surface roughness of the rake and flank faces are indicated although this roughness affect the cutting force dramatically.
- Only one cutting speed is used. Although three tools made of different tool materials (judging by their application recommendations), having considerably different coating and cutting geometry used in the tests, the cutting speed and feed per tooth are chosen to be THE SAME for these tools with no justification. The authors should be aware that each of the tested tool may have its own optimal cutting conditions in terms of minimizing its wear.
Reviewer 3 Report
The paper deals with the Wear study of coated mills during circumferential milling of
carbon fiber reinforced composites and their influence on the
sustainable quality of the machined surface.
According to the reviewer, the paper is not worth publishing at Coatings Journal.
Comment 1
Line 5
Tomas Knapek*,
1 or 2 or 1 and 2?
The authors must add the affiliation of the author Tomas Knapek.
Comment 2
Lines 15 - 16
Three types of instruments
The authors must check if the word "instruments" is right (machine tools?).
Comment 3
The authors must add the results in the abstract.
Comment 4
Lines 45 and 48
The authors must format the paper according to the journal's instructions.
The authors first present the ref. [10] and then the ref. [3].
References: References must be numbered in order of appearance in the text
(including table captions and figure legends) and listed individually at the end of the manuscript.
Line 61
The authors must format the paper according to the journal's instructions.
[4], [5].
The authors must replace
[4-5].
Lines 63 - 64
[2], [5],
[6], [7], [8], [9], [10], etc.
The authors must replace (delete the "etc")
[2, 5 - 10].
Line 66
with CFRP, etc. Research
The authors must replace (delete the "etc")
with CFRP. Research
Table 01 or Figure 01
The authors must replace
Table 1 or Figure 1
Line 164
publication [12], [13], [14], [15].
The authors must replace
publication [12 - 15].
Line 484
[18], [19], [20], [21], [22], [23], [25], [26], etc.
The authors must replace (delete the "etc")
[18 - 26].
Comment 5
Line 80
The authors must give more details for the Glossy laminated 3K CFRP plates (from supplier or author's experiments?).
Line 84
The authors must give more details for the Table 1 (the values are from supplier or author's experiments?)
Comment 6
Figure 1
The authors must explain in the middle of the Figure the distance between the layers are not constant.
Comment 7
Lines 93 - 95
Laminated CFRP plates with a thickness of 1 and 3 mm were cut in the form of plates
with dimensions of 400x250 mm. The boards were adjusted/cut to sample dimensions of
200x250 mm using a band saw.
While the cutting edge of the band saw is constant, there is not understandable how the authors cut the initial 400x250 mm to final (2) 200x250 mm.
Comment 8
Table 2
The authors must insert data for the machine tool's substrate (not only data for the coatings).
Comment 9
Lines 125 - 126 and equations (1) - (2)
The author's analysis is not comprehensible.
Comment 10
The authors must add a Figure with the Machine and the experimental device.
Comment 11
Line 140
The authors must explain how the value of 165 m occurs.
Comment 12
Figures 6, 7, 8 and 9 (a)
There are arroys without dimensions.
Comment 13
Figure 10
In the Figure 10 : 284.550 μm
In the text for Figure 10 : 74.70 ± 0.16 µm.
The authors must explain the difference.
Comment 14
According to the journal's instructions there is no Graphs.
Comment 15
The authors must explain:
Since the wear is increasing, how is it explained that the force Fx decreases (in some cases)?
Since the wear is increasing, how is it explained that the Roughness decreases (in some cases)?
Comment 16
Major Problem:
Lines 461 - 472
The authors must explain how statistically the standard deviation is contant (0.10 μm).
Comment 17
Line 545
Delete the extra [12].
Comment 18
The authors must comment in the text the Ref. [16] and [17].
Comment 19
Increase the number of the reference papers including (primarily) from MDPI journals.
The authors use 0 paper Coatings / 0 paper from MDPI journals / 26 papers from journals (References)
Τhe number for papers from MDPI journals
is considered insufficient (in reviewer's opinion).
Comment 20
Changes the References section format.
According to the journal's instructions:
1. Author 1, A.B.; Author 2, C.D. Title of the article. Abbreviated Journal Name Year, Volume, page range.
The authors must correct the References according to the journal's instructions.
Reviewer 4 Report
The paper is well written and organised, the scientific approach is good and the presented results look legit.
There are only a few small observations.
Page 2, lines 62-63: the expression "in the given area" is repeated twice in the same sentence.
There are a few places in the text where the authors make references to certain graphs (page 11, line 297; page 13, line 349; page 19, line 490) instead of the corresponding figures, which makes it difficult to follow the explanations.
In the text, the authors use the tool types when discussing the results whereas in the corresponding figures they use the coating types. These differences may confuse the reader and raise the difficulty of text understanding.
The first sentence of the Discussion section (page 17, lines 418-421) is rather unclear.
Author Response
Thank you for your report.
The graphs have been replaced with figures.
''in the given area'' was corrected
The first sentence of discussion was overwritten.
Reviewer 5 Report
The authors presented research of circumferential milling of CFRP sheets with a focus on cutting forces and tool flank face wear, including their effect on the machined surface structure, roughness, and topography of the laminate. The main objective of the study was to investigate the feasibility of applying conventional coated tools. The study results confirm the possibility of using conventional tools for machining CFRP.
The paper is interesting, but I have an irresistible feeling that it should be revised by an English native speaker.
Weak
1. The paper's biggest weakness is its shallow analysis of the state of the issue, or rather the lack thereof. The wholesale citation of literature sources in the form of [6 - 15] is unacceptable. Each cited source should be described in a few sentences. If the sources are thematically similar then they can be grouped, but not more than 2 to 3.
2. In addition, the sources cited are quite old. Only 5 out of 29 are from the last 3 years. This should be changed and the analysis of the state of the issue should be expanded, as the paper in this form is more suitable for a conference than for a reputable journal.
Noticed errors
1. In the abstract, the authors provide acronyms that should be described beforehand like CFRP, or DLC.
2. Table 1, Table 3 Aliquots of the unit of length - cm - are acceptable but not recommended in engineering sciences as opposed to tailoring. The units kg/m3 should be used
3. Figure 2 is poor quality. It should be corrected.
4. Bad pagination of 7/8, 8/9 pages
5. The work should be expanded into conclusions for further research
Small errors
1. Put a space between the magnitude and the value, for example line 111: Is 55HRC, should be 55 HRC. Applies to the entire work.
2. In the equation (1) and (2), is mm a variable or a unit? If unit then it cannot be in italics.
Author Response
Thank you for your report.
The references have been redesigned and additional information has been added to the introduction.
a few newer resources has been added
CFRP, DLC abbreviations were explained
Figure 2 is now in better quality
the work was be expanded into conlusion for further research
equation 1 and 2 was corrected
space between the magnitude and value was added.
Round 2
Reviewer 1 Report
The paper has been revised according to the comments. However, some issues still lack. For example, the selection of cutting condition, the tool wear traces under different conditions. This is not rigirous for tool wear study of milling process. Please check.
Reviewer 2 Report
What I do not like is the following author's responce:
"Additional information on the material composition has been added. The hardness and thickness of the coatings were supplied by the manufacturer. The manufacturer was not able to provide further details on the geometry and methods of determining the parameters mentioned. We are still communicating with the manufacturer about more information."
You can measure the parameters using a standard tool measuring machine as, for example, Zoller Genius 3.
Reviewer 3 Report
Comment 1
The abstract is 420 words.
Abstract: A single paragraph of about 200 words maximum.
Comment 2
Line 73
15]. on the investigated issue,
The authors must replace
15]. On the investigated issue,
Exteded text editing.
Comment 3 (previous 4)
Table 01 or Figure 01
The authors must replace
Table 1 or Figure 1
Corrected
But the authors did not the corrections (for example Line 83).
Comment 4
Table 3
Finishing cutter ECSSF: H10 OR H105?
The authors must give more details for the machine tool.
Comment 5
Figure 3
The authors must add a Figure with the machine tool and the dynamometer.
Comment 6 (previous comment 9)
Lines 125 - 126 and equations (1) - (2)
The author's analysis is not comprehensible.
Corrected notes and scheme for equations.
The authors mut explain how calculated the total cutting depth.
Comment 7
Lines 161 - 162
While the cutting distance per path is 200mm,
the authors must expalin "Until the traveled path of 3300m (20x 165 m)"
Comment 8
Line 195
It is not so good to use the word "we".
The authors must rephrase.
Comment 9 (previous comment 14)
Comment 14
According to the journal's instructions there is no Graphs.
There are Graphs 01-10.
The authors must format the paper according to the Journal's instructions.
Comment 10 (previous comment 20)
Changes the References section format.
According to the journal's instructions:
1. Author 1, A.B.; Author 2, C.D. Title of the article. Abbreviated Journal Name Year,Volume, page range.
The authors must correct the References according to the journal's instructions.
Corrected
But the authors did not the corrections.
Comment 11 (previous comment 13)
Comment 13
Figure 10
In the Figure 10 : 284.550 μm
In the text for Figure 10 : 74.70 ± 0.16 μm.
The authors must explain the difference.
Wrong picture inserted. For the purpose of the experiment, tool wear was taken into
account - 74.70 ± 0.16 μm
But the authors did not the corrections.
Comment 12 (previous comment 15 and 16)
The authors must increase the quality of their answer.
